# Identification of compounds that cause axonal dieback without cytotoxicity in dorsal root ganglia explants and intervertebral disc cells with potential to treat pain via denervation

Fei San Lee[1], Uyen N. Nguyen[1], Eliza J. Munns[2], Rebecca A. Wachs[1]*

1 Department of Biological Systems Engineering, University of Nebraska-Lincoln, Lincoln, Nebraska United States of America, 2 Department of Electrical, Computer, and Biomedical Engineering, Union College, Schenectady, New York, United States of America

* rebecca.wachs@unl.edu

**Data Availability Statement:** All relevant data are within the paper and its supporting information files.

## Abstract

Low back pain, knee osteoarthritis, and cancer patients suffer from chronic pain. Aberrant nerve growth into intervertebral disc, knee, and tumors, are common pathologies that lead to these chronic pain conditions. Axonal dieback induced by capsaicin (Caps) denervation has been FDA-approved to treat painful neuropathies and knee osteoarthritis but with short-term efficacy and discomfort. Herein, we propose to evaluate pyridoxine (Pyr), vincristine sulfate (Vcr) and ionomycin (Imy) as axonal dieback compounds for denervation with potential to alleviate pain. Previous literature suggests Pyr, Vcr, and Imy can cause undesired axonal degeneration, but no previous work has evaluated axonal dieback and cytotoxicity on adult rat dorsal root ganglia (DRG) explants. Thus, we performed axonal dieback screening using adult rat DRG explants in vitro with Caps as a positive control and assessed cytotoxicity. Imy inhibited axonal outgrowth and slowed axonal dieback, while Pyr and Vcr at high concentrations produced significant reduction in axon length and robust axonal dieback within three days. DRGs treated with Caps, Vcr, or Imy had increased DRG cytotoxicity compared to matched controls, but overall cytotoxicity was minimal and at least 88% lower compared to lysed DRGs. Pyr did not lead to any DRG cytotoxicity. Further, neither Pyr nor Vcr triggered intervertebral disc cell death or affected cellular metabolic activity after three days of incubation in vitro. Overall, our findings suggest Pyr and Vcr are not toxic to DRGs and intervertebral disc cells, and there is potential for repurposing these compounds for axonal dieback compounds to cause local denervation and alleviate pain.

## Introduction

Around the world, 18.1 million people have cancer [1], 540 million people have low back pain (LBP) [2] and 17.1 million people suffer from hip and knee osteoarthritis (OA) [3]. These

**Funding:** RAW received the NSF CAREER award (NSF CAREER1846857) from the National Science Foundation (NSF) which provided funds to conduct this research. RAW, FSL, UNN received funding by NSF CAREER1846857. EJM received funding from NSF Research Experiences for Undergraduates (REU) program EEC 2050587 as a part of this work. The funders had no role in study design, data collection and analysis, decision to publish, or preparation of the manuscript.

**Competing interests:** The authors have declared that no competing interests exist.

diseases severely impact a person's quality of life. At least 95.6% of cancer patients suffer from pain during disease progression [4]. Globally, LBP and knee OA are ranked as the number one and 13th leading cause for years lived with disability, respectively [5]. Aberrant nerve sprouting is a common pathological feature leading to cancer pain [6–8], knee OA [9–11] and disc-associated LBP [12–15]. Sensory neurons that grow into tumors, neuromas, bones, joints, or intervertebral discs can be sensitized by inflammatory or mechanical stimuli leading to pain [7, 12, 16]. Despite the known etiologies of the pain, initial treatments rely on pain relief medications using non-steroidal anti-inflammatory drugs (NSAIDs), and opioids or adjuvants [17], which often do not offer long lasting efficacy [18]. Further, prolonged use of NSAIDs can potentially lead to life threatening complications including gastrointestinal bleeding and stomach ulcers [19–21] and opioid use may lead to misuse, addiction, and death due to overdose [22, 23]. For cancer patients, treatments for the pain are palliative and only rely on oral morphine or adjuvants [24], despite known risks. For patients with end stage knee OA and intractable chronic LBP, knee arthroplasty or spinal fusion surgeries are often performed, which involve risks of surgery complications and overall, poor long-term improvements [25–27].

An interventional therapy that targets aberrant nerve growth at the source of pain without adverse side effects is needed. One strategy is to remove aberrant nerves innervating the injured tissue using axonal dieback compounds. This process is called denervation. The only FDA-approved drug in the market that treats pain via denervation is the Qutenza 8% capsaicin (Caps) dermal patch, which is approved for use for post-herpetic neuralgia [28], HIV-associated neuropathy [29], and diabetic peripheral neuropathy [30]. A high dose of Caps induces a large calcium influx which activates calpain, an enzyme that catalyzes cytoskeletal breakdown, leading to nerve degeneration [31]. The disappearance of nerve fiber terminals in the epidermis after Caps dermal patch application correlates with the loss of sensation and pain alleviation [32, 33]. However, high concentration Caps patches commonly report adverse events such as coughing [34, 35], skin irritation [36], and intense burning pain hours after application [37, 38]. As a result of skin reactions, patients withdraw from Caps medication [39]. Preliminary in vivo cancer studies demonstrated systemic denervation using Caps may lead to side effects tumor metastasis [40] and aggravation of pulmonary inflammation [41]. Further, the pain relief effects from Caps is only short-term and require repeated administration due to subsequent reinnervation [42]. Alleviation of knee pain after intraarticular Caps injection only lasted about 18 weeks in a human clinical study [37]. Denervation is a promising approach for relieving pain due to aberrant nerve sprouting but there is a need to identify other axonal degeneration compounds that have the potential for reduced side effects.

Axonal dieback is a process of neurodegeneration whereby degeneration is initiated at the distal axon and progresses proximally leading to shortening of the axon towards the soma [43, 44]. Commercially available drugs with undesirable nerve degeneration side effects may hold great potential to be repurposed for inducing local denervation with potential analgesia similar to Caps. Drugs that have shown off-target peripheral neurodegenerative effects include pyridoxine (Pyr) [45–48], vincristine sulfate (Vcr) [49–58], and ionomycin (Imy) [59, 60].

Pyr is a derivative of vitamin B6, and a co-factor involved in enzymatic pathways for amino acid metabolism [61, 62]. Pyr is used to treat conditions such as seizures [63], premenstrual syndrome [64], and peripheral neuropathy [65]. However, at high doses and prolonged oral supplementation, Pyr has been shown to lead to neuropathic symptoms such as progressive burning pain, numbness, tingling and muscle weakness concurrent with evidence of nerve damage [45, 66]. High dose Pyr resulted in axonal degeneration and vacuolation when injected subcutaneously in rodents [47, 48] and dogs [67]. One study observed dose-dependent alleviation of thermal pain, but no effect on mechanical pain or normal pain sensation after repeated administration of Pyr [68]. No previous works have tested effects of Pyr on neurodegeneration

and cytotoxicity on rat dorsal root ganglia (DRG) neurons in vitro. Further work to determine a safe and effective dose to induce Pyr-induced axonal degeneration locally without FRG cytotoxicity is needed to potentially repurpose Pyr for denervation.

Vcr is a vinca alkaloid widely used as an anti-cancer drug [69] because it specifically binds to β-tubulin and prevents microtubule formation in mitotic spindles, thereby arresting cell division [69]. β-tubulin is also a key cytoskeletal component for anterograde and retrograde transport in neurons [70, 71]. Therefore, Vcr also impairs axonal transport [52, 72] which leads to axonal degeneration [50, 51, 58, 72]. 96% of chemotherapy patients develop peripheral neuropathy such as numbness and tingling due to axonal degeneration caused by systemic administration of Vcr [54, 73–75]. In mouse models, locally injecting Vcr into the hind paw resulted in transient mechanical hypersensitivity but later transitioned into long-term hypo-sensitivity or loss of sensation [76]. Treatment of dissociated DRG neurons from rodents demonstrated neurite fragmentation and loss of neurite processes [50, 77]; however, the DRG dissociation process removes support cells from culture that may be necessary for axonal dieback. Vcr is a promising drug for axonal dieback, but dose-dependent effects and cytotoxicity have not been previously studied in adult DRG explants and nucleus pulposus cells.

Imy is a membrane permeable ionophore that selectively binds to calcium ions in a 1:1 ratio and used as an antibiotic to treat infections [78]. This drug induces calcium ion influx [59] and discharges endoplasmic reticulum calcium store [79] which can lead to neurodegeneration in rodents [80–82]. When applying Imy onto sciatic nerves, segmental demyelination was observed, with up to 95% of the fibers exhibiting large nerve lesions [60]. Further, treatment of mouse neuroblastoma cells with Imy resulted in axon degeneration and cell death [59]. Despite evidence of axon degeneration and demyelination, there is no previous work studying the cytotoxicity effects of direct application of Imy on axonal dieback in DRG explants.

The effects of Pyr, Vcr and Imy induced nerve degeneration or neuropathic pain have mostly been observed after systemic delivery. There is a need to characterize the axonal degeneration effects when applying these drugs locally on nerve fiber endings to determine their potential efficacy to alleviate pain by local denervation without off-target effects. In addition, neuronal cell death is associated with pain [83]. Hence, the objective of this research is to identify drugs that have axonal dieback properties similar to Caps while minimizing cytotoxicity, using adult rat DRG explants cultured in 3D hydrogels. We screened axonal dieback compounds on male and female rat DRGs to determine how drug doses may impact axonal dieback and cytotoxicity. To further explore the repurposing capability of these drugs for intervertebral disc denervation, we evaluated cytocompatibility with nucleus pulposus (NP) cells, the resident chondrocyte-like cells of the disc. We quantified cell viability and metabolic activity after direct exposure of Pyr and Vcr on human NP cells in vitro over a range of concentrations. This research describes a method for screening potential axonal dieback compounds and provides a foundation for future development of denervation products.

## Methods

### Treatment and control solution preparation

Drug stock solutions were prepared in either 1X phosphate buffered saline (PBS) or ethanol (EtOH) solvent then diluted in incomplete DRG media: Neurobasal-A media (Thermo Fisher Scientific, 10888022) supplemented with 10% Fetal Bovine Serum (FBS) (Fisher Scientific, 26-140-079), 1% GlutaMax (Thermo Fisher Scientific, 35050–061), 1% Penicillin Streptomycin (Fisher Scientific, 15-140-122) and 2% B-27 (Thermo Fisher Scientific, 17-504-044) until the desired final concentrations in gels are achieved for axonal dieback screening. Incomplete DRG media is not supplemented with Nerve Growth Factor (NGF) and used during axonal

dieback screening phase to avoid confounds since NGF has neuroprotective effects from Vcr-induced axon degeneration [84].

To prepare the stock solutions, capsaicin (Sigma-Aldrich, 360379-1G, MW: 305.41 g/mol) was weighed and resuspended in EtOH. Based on the concentration and surface area used in 8% topical Caps dermal patch [28], the concentration used was scaled down to the surface area of the hydrogel platform, and thus was diluted to 3.79 mM Caps as a positive control for axonal degeneration. Pyridoxine hydrochloride (Sigma-Aldrich, P6280-10G, MW: 205364 g/mol) and vincristine sulfate (Sigma-Aldrich, V0400000, MW: 923.04 g/mol) were dissolved in 1X PBS. Pyr stock solution was vortexed until the drug dissolved then stored at 37°C until ready for dilution to avoid precipitation. Pyr stock solution was diluted to reach 50 μM, 500 μM and 1 mM Pyr. Vcr stock solution was diluted to achieve 10 nM, 100 nM, 200 nM and 500 nM Vcr. Ionomycin (Cayman, 10004974, MW: 709 g/mol), supplied as 1 mg/mL in EtOH, was diluted directly in incomplete media to achieve 500 and 770 nM Imy. To test Imy for axonal dieback, a maximum concentration of 770 nM Imy was chosen to minimize the final EtOH concentration in treatment media solution to 0.055% v/v EtOH. Control solutions consisted of media and solvents (0.055% v/v EtOH or 1X PBS) without any compounds. 0.055% v/v EtOH is the negative control for Imy and Caps, while 0.055% v/v 1X PBS was the negative control for Pyr and Vcr. All drug stock and media treatment solutions were freshly prepared on the same day of experiment. Drug and solvent concentrations reported are final concentrations in the wells accounting for the total volume in each well (550 μL) including the media solution and gel.

## DRG explant hydrogel culture platform for axonal dieback compound screening

**Adult rat DRG explant harvest and culture.** The DRG is a cluster of cell bodies of sensory neurons that connect the peripheral nervous system to the central nervous system. A subset of sensory neurons called nociceptors are responsible for detecting noxious stimuli and signaling pain [85]. In vitro culture of adult rat DRG explants are a relevant cell type compared to dissociated DRGs or underdeveloped neurons from neonates and embryos because they are mature and contain support cells necessary for growth and maintenance [86–88]. Our lab has established a DRG explant culture platform with robust axon growth in three-dimensional hydrogels consisting of collagen type I, laminin I and methacrylated hyaluronic acid (MAHA) [89, 90]. All animal experiments were conducted in accordance with the Guide for the Care and Use of Laboratory Animals approved through the University of Nebraska-Lincoln's Institutional Animal Care and Use Committee (IACUC) and adherence to the Animal Research: Reporting of In Vivo Experiments (ARRIVE) guidelines. Adult male and female Sprague Dawley rats (Charles River) aged 10 to 13 weeks were humanely euthanized via $CO_2$ inhalation followed by bilateral thoracotomy. The procedure to harvest adult rat DRGs has been previously described by Piening et al [89]. Briefly, bilateral T8-L6 DRGs were surgically removed, and surrounding connective tissues were cleaned and trimmed under a Zeiss stereomicroscope (Carl Zeiss Microscopy, Inc.). DRGs were cut into roughly 0.6 mm-diameter pieces then one piece of DRG was embedded into 250 μL hydrogel solution in a 48-well plate. Hydrogel solutions were prepared by mixing 0.75 mg/mL laminin I (R&D Systems, 34446-005-01), 1.25 mg/mL MAHA (Sigma-Aldrich, 53747-10G) and 4.5 mg/mL sodium hydroxide neutralized collagen type I (IBIDI, 50205) in 1X PBS (pH 7.3–7.5). MAHA was prepared via methacrylation using methacrylic anhydride as previously described by Seidlits et al. [91] and used previously by our lab [89, 90]. The degree of MAHA methacrylation used in this study ranged between 85 to 115%. Up until embedding, DRGs were processed in cold incomplete DRG media. The hydrogel solution with the embedded DRG explant was thermally cross-linked for 30 minutes

at 37˚C then UV-photo crosslinked for 90 seconds under a 10-19W UV lamp (UVP B 100-AP, Analytik) [90]. DRGs were maintained in culture at 37˚C, 5% $CO_2$ with 300 μL complete DRG media, which is incomplete DRG media supplemented with 50 ng/mL NGF (R&D Systems, 556-NG-100) to induce neurite sprouting. Media was fully replaced after one hour and 33.3% v/v of the media was partially replaced every three days. During the axonal outgrowth phase, traced axon lengths and maximum radial distance of male rat DRGs (n = 28) and female rat DRGs (n = 35) were measured on days 7, 14 and 20. Axon growth rates were calculated as the percent difference of traced axon length on day 14 to day 7 and on day 20 to day 14. Traced axon length and maximum radial axon distance significantly increased over time between days 7 and 20 (p<0.0001) in hydrogel culture platform with no significant sex differences in axon length, maximum radial distance, and axon growth rate between male and female rat DRGs (**S1A and S1B Fig**). DRG axon outgrowth rate was significantly higher between days 7 to 14 compared to between days 14 to 20 suggesting DRGs active initial axon outgrowth phase until day 14 then growth slows down after day 20 in vitro (**S1C Fig**). Hence, we choose to test axonal dieback screening after DRGs have established axon outgrowth between days 16 to 21. DRGs without substantial axon extension (less than five axons per DRG) by day 16 to 21 in culture were excluded from axonal dieback screening experiments.

**Axonal dieback screening in DRG explants.** DRGs exhibited long axon extensions in vitro at which time compounds of interest were added: Caps (3.79 mM), Pyr (50 μM, 500 μM, 1mM), Vcr (1, 10, 100, 200, 500 nM) or Imy (500, 770 nM). 500 nM Imy and 10 nM Vcr were excluded from female rat DRG experiments while 500 nM Vcr and 1 mM Pyr were excluded from male rat DRG experiments due to insufficient DRG replicates with axon outgrowth. DRG media was removed from wells and transferred into a 96-well plate for pre-treat LDH cytotoxicity assay (see Methods below). 300 μL of media treatment solutions or solvent control solutions were added to wells. DRGs were incubated with treatment solutions for three days, and brightfield images at 4X magnification were taken using BioTek Cytation1 Plate Imager (Agilent) before treatment (day -1) and on days 1, 2 and 3 after treatment. DRGs from adult male and female rats were tested to determine if sex can affect axonal dieback. In total, 12 to 15 DRG replicates per group by sex harvested from n = 3 male and n = 3 female rats were tested.

## DRG axon length analysis

Changes in axon length were used to characterize axonal dieback. Four to 12 continuous axon paths were randomly selected per DRG and manually traced at each timepoint after compounds of interest were added using SimpleNeuriteTracer (SNT) plugin in ImageJ (FIJI) [92]. To ensure the same DRG axons were traced before and after treatment, the brightfield DRG images collected pre-treatment and on Day 1, Day 2 and Day 3 post-treatment were compiled into a montage and the same axons were traced in the image sequence. To identify axonal dieback, the length of each axon path traced on days 1, 2 and 3 post treatment were normalized to axon length pre-treatment (day -1) and reported as "axon length ratio" as shown in Eq 1. The axon length ratio represents the fold change in axon length over time. The average axon length ratio from each DRG was calculated and values above 1.0 indicate axon growth and values below 1.0 indicate axonal dieback. Maximum radial distance was measured using the "straight line tool" and "measure tool" in Image J (FIJI) [93] by drawing a straight perpendicular from the longest axon to the nearest point perpendicular to the soma boundary.

$$Axon\ length\ ratio = \frac{Axon\ length\ ratio\ post-treat}{Axon\ length\ ratio\ pre-treat} \tag{1}$$

## DRG cytotoxicity assessment using LDH assay

Lactate dehydrogenase (LDH) is a cytosolic enzyme present in all cells. Dissociated rat DRG neurons release LDH during tunicamycin-induced apoptosis [94]. Hence, LDH released into media supernatant is an indicator of cells undergoing apoptosis or damage to cell membrane. An ideal axonal dieback compound should cause axonal degeneration without neurotoxicity. was assessed on day 3 post-treat and normalized to pre-treat levels to account for variability in DRG size and growth and calculated as a percentage of cytotoxicity to lysed DRGs (positive control). 50 μL of pre-treatment and post-treatment media were aliquoted in duplicates into a 96-well plate. LDH activity in the media post-treat was assessed using the CyQUANT$^{TM}$ LDH Cytotoxicity Assay Kit (Thermo Fisher Scientific, C20301) according to the manufacturer's instructions. Briefly, 50 μL of Substrate Mix was added to each well and then incubated at room temperature for 30 minutes protected from light. 50 μL of Stop solution was added to each well and the absorbance at 490 nm and 680 nm was measured using a microplate reader (Agilent). LDH activity of each DRG was calculated as the difference between absorbance at 490 nm after subtracting the background absorbance at 680 nm. As a positive control, 13 female rat DRGs and 15 male rat DRGs were treated for three days with Lysis buffer solution, provided in the kit, diluted 1:5.45 in DRG media. The percentage of cytotoxicity was calculated following Eq 2. Pre-treat levels of LDH activity were subtracted from post-treat LDH activity in each DRG to remove baseline LDH activity then percentage of cytotoxicity compared to positive control lysed DRG LDH activity was calculated. Sex is a biological factor that can influence neurotoxicity [95]. Thus, percentage of cytotoxicity of total 11 to 14 DRG replicates per group by sex from n = 3 male and n = 3 female rats were tested.

$$Cytotoxicity\ (\%) = \frac{(\ Post\ treat\ LDH\ activity\ ) - (\ Pre\ treat\ LDH\ activity\ )}{(\ Maximum\ LDH\ activity\ from\ lysed\ DRGs\ ) - (\ Pre\ treat\ LDH\ activity\ )} \times 100\% \quad (2)$$

## Nucleus pulposus cell cytotoxicity evaluation

**Human nucleus pulposus in vitro cell culture and drug treatment.** Nucleus pulposus (NP) cells are native matrix-producing cells at the core of intervertebral discs. To repurpose axonal dieback compounds to denervate aberrant nerve fibers in discs related to chronic LBP, the cytocompatibility of these compounds with NP cells must be evaluated. Human NP cells (Sciencell, 4800) from donor 1 (Lot #5954, male) and donor 2 (Lot# 21579, female) were cultured according to previously established protocol in our lab adapted from manufacturer's instructions [96]. Briefly, passage three cells from both donors were expanded in Poly-L-lysine (PLL) (Sciencell, 413) coated T75 flasks. Cells were then lifted and reseeded at 5,000 cells/cm$^2$ onto two PLL-coated 48-well plates (Greiner Bio-One, 677180) and allowed to attach overnight before treating with Pyr or Vcr. One plate was used for LIVE/DEAD staining and the other plate was used for AlamarBlue assay as described below. Complete NP cell media (Sciencell, 4801) containing basal media supplemented with 1% penicillin-streptomycin, 1% growth supplement and 2% FBS were used for maintaining cells in culture. To mimic the hypoxic disc environment, NP cells were cultured in a hypoxia chamber (Stem Cell Technologies, 27310) with 5% CO$_2$, 3.5% O$_2$, 91.5% N$_2$ (Airgas) during the entire culture and treatment incubation period.

Pyr stock solution was diluted in complete NP media to achieve final concentrations of 500 μM, 1 mM and 2 mM Pyr. 16.9 μg/mL and 169 μg/mL Vcr stock solutions were prepared

in 1X PBS then diluted in complete NP media to achieve final concentrations of 10, 25, 50, 100 nM and 200, 500 nM, respectively. Cells were incubated with 250 μL of final treatment solutions in duplicate wells per donor for three days. Cells in negative control group were incubated with 250 μL of complete NP media with 0.55% v/v 1X PBS as solvent control. On day 3 post-treatment, brightfield images at 4X magnification were taken using a BioTek Cytation1 Plate Imager (Agilent) to verify cell proliferation and observe cell morphology. Three experimental replicates were performed for each cell donor.

**Quantification of viable cells.** On day 3 of incubation with drug, media treatment solutions were removed from each well and cells were carefully rinsed with 200 μL of sterile 1X PBS to remove excess media. NP cells were stained with 150 μL of 2 μM Calcein AM and 6 μM EtHD-1 solution from LIVE/DEAD Viability/Cytotoxicity Kit (Thermo Fisher Scientific, L3224) for 30 minutes at room temperature protected from light. After staining, cells were gently rinsed with 200 μL sterile 1X PBS to remove excess staining solution. Excitation and emission wavelengths for live cell fluorescence is 495 and 515 nm and dead cell fluorescence is 495 and 635 nm. Using a BioTek Cytation1 Plate Imager (Agilent), GFP filter cube (Agilent, 1225101, ex: 451.5–486.5 nm, em: 505.5–544.5 nm) and RFP filter cubes (Agilent, 1225103, ex: 511-551nm, em: 573–613 nm), fluorescent images at 4X magnification of four regions per well were taken. Gen5 v3.11 software (Agilent) was used to count green (live) and orange (dead) cells in fluorescent images. Dead NP cell staining was verified and evident by positive EtHD-1 orange staining in kill control wells where cells were treated with 70% EtOH for 10 minutes (S4 Fig). Percentage of live cells were calculated by calculating the percentage of live cells over total cell counts as shown in Eq 3.

$$Cell\ viability\ (\%) = \frac{Live\ cell\ counts}{Live\ cell\ counts + Dead\ cell\ counts} \times 100\% \quad (3)$$

**Cell metabolic activity measurements.** On day 3 of incubation with drugs, 25 μL (10% v/v of media volume) of AlamarBlue HS cell viability reagent (Thermo Fisher Scientific, A50100) were added into each well. Cells were incubated with the AlamarBlue reagent for 2 hours at 37°C under hypoxia then 60 μL aliquots of cell media were removed in duplicates into a 96-well plate. The absorbance at 570 and 600 nm were read using a microplate reader (Agilent) and calculations for AlamarBlue reduction were calculated according to the manufacturer's protocol. To account for different cell confluency between donors and groups, AlamarBlue reduction was normalized to estimated total number of cells per well obtained from LIVE/DEAD staining. Cellular metabolic activity in treated groups was reported as a ratio of normalized AlamarBlue reduction to PBS control group.

## Statistical analysis

All statistical analyses were conducted using GraphPad Prism Version 10.0.0. Normality of data was assessed using Shapiro-Wilk test with p-value set to 0.05. Traced axon length and maximum radial distance data during outgrowth phase was normally distributed and a 2-way ANOVA with Sidak's post-hoc test was used to analyze sex differences. DRG axon growth rate data during outgrowth phase and axon length ratio data from axonal dieback screening failed the normality test. As a result, axon length ratio is reported as median and interquartile ranges of DRG replicates. Mann-Whitney test was used to analyze sex differences in axon length ratio for each drug and DRG axon growth rate. The multiple Wilcoxon test was used to compare axon length ratio between days. The axon length ratio for each compound was compared to respective solvent controls at each timepoint using the Mann-Whitney test when comparing

two groups and Kruskal-Wallis test with Dunn's multiple comparison test when comparing more than two groups. To identify sex differences in DRG cytotoxicity for each group, Multiple Mann-Whitney test was used. Percentage of cytotoxicity were comparable between male and female DRG explants, and no sex differences were detected (**S3 Fig**). Hence, cytotoxicity levels from female and male rat DRG explants were pooled for group comparison analysis. DRG cytotoxicity pooled data for all groups except 1 nM Vcr and lysed DRGs were normally distributed. Unpaired two-tailed t-test for two-group comparison and ordinary one-way ANOVA for multiple group comparison were used to analyze DRG cytotoxicity for Caps, Pyr and Imy in comparison to matched solvent control. Kruskal-Wallis with Dunn's multiple comparison test was used to analyze DRG cytotoxicity for Vcr groups in comparison to PBS control and all groups in comparison to lysed DRGs. DRG cytotoxicity was reported as mean and standard deviation between DRG replicates. Percentage of viable NP cells did not pass the normality test, so control and all treatment groups from each donor were analyzed using Kruskal-Wallis test with Dunn's multiple comparison test and donor effects were analyzed using Wilcoxon's matched-pair test. Normalized AlamarBlue reduction data passed the normality test and were analyzed using 2-way ANOVA with Tukey's post-hoc test to compare all treatment groups for each donor. Percentage of viable NP cells and normalized AlamarBlue reduction were reported as mean and standard deviation. Exact significant p-values above 0.001 were reported in text and symbols on graphs describe all significant differences detected with $p < 0.05$. Data files for axon length and cytotoxicity can be found in **S1 Data** and data files for nucleus pulposus cell viability and metabolic activity can be found in **S2 Data**.

## Results

### Capsaicin induced axonal dieback in both male and female DRG explants in a sex-dependent manner and is neurotoxic

DRG explants treated with 3.79 mM Caps exhibited axons retracting toward the soma starting on day 1 and continuing to progress up to day 3 (**Figs 1 and 2 and S3 Video**), whereas solvent controls continue to grow (**S1 and S2 Video**). The axon length ratio from 3.79 mM Caps-treated DRGs significantly decreased over time in male and female rat DRGs ($p < 0.001$) (**Fig 3A**). Female rat DRGs had significantly faster axonal dieback on day 2 compared to male rat DRGS after treatment with Caps, but both male and female DRGs reach a similar axon length ratio by day 3 (**Fig 3A**). The median of axon length ratio in male rat DRGs was 0.657 (0.359–0.750) while female rat DRGs was 0.453 (0.270–0.624) on day 3 post-treat with Caps (**Fig 3A**). This signifies that less than 65.7% and 45.3% of initial axon length was remaining in male and female DRG explants after the 3-day incubation with Caps. Since Caps is widely known to cause axonal dieback at high doses, a significant reduction in axon length ratio after Caps treatment validates the use of our in vitro DRG explant hydrogel culture platform as a tool to screen for other axonal dieback compounds. To identify novel axonal dieback compounds that may be able to alleviate pain like capsaicin, an axon length ratio of 0.45 was used as a minimum criterion for selection. The percentage of cytotoxicity in Caps-treated DRGs compared to respective EtOH control DRGs showed a significant increase in cytotoxicity levels after 3-day incubation suggesting Caps may be slightly neurotoxic (**Fig 3B**). However, DRG cytotoxicity is minimal as the percentage of cytotoxicity was 90.2% lower than in lysed DRGs ($p < 0.001$) (**Table 1**).

Mean and standard deviation of percentage of DRG cytotoxicity from male and female rat DRGs after treatment with either vehicle control or compound solution. The percentage of cytotoxicity did not differ significantly between male and female rat DRGs and were pooled

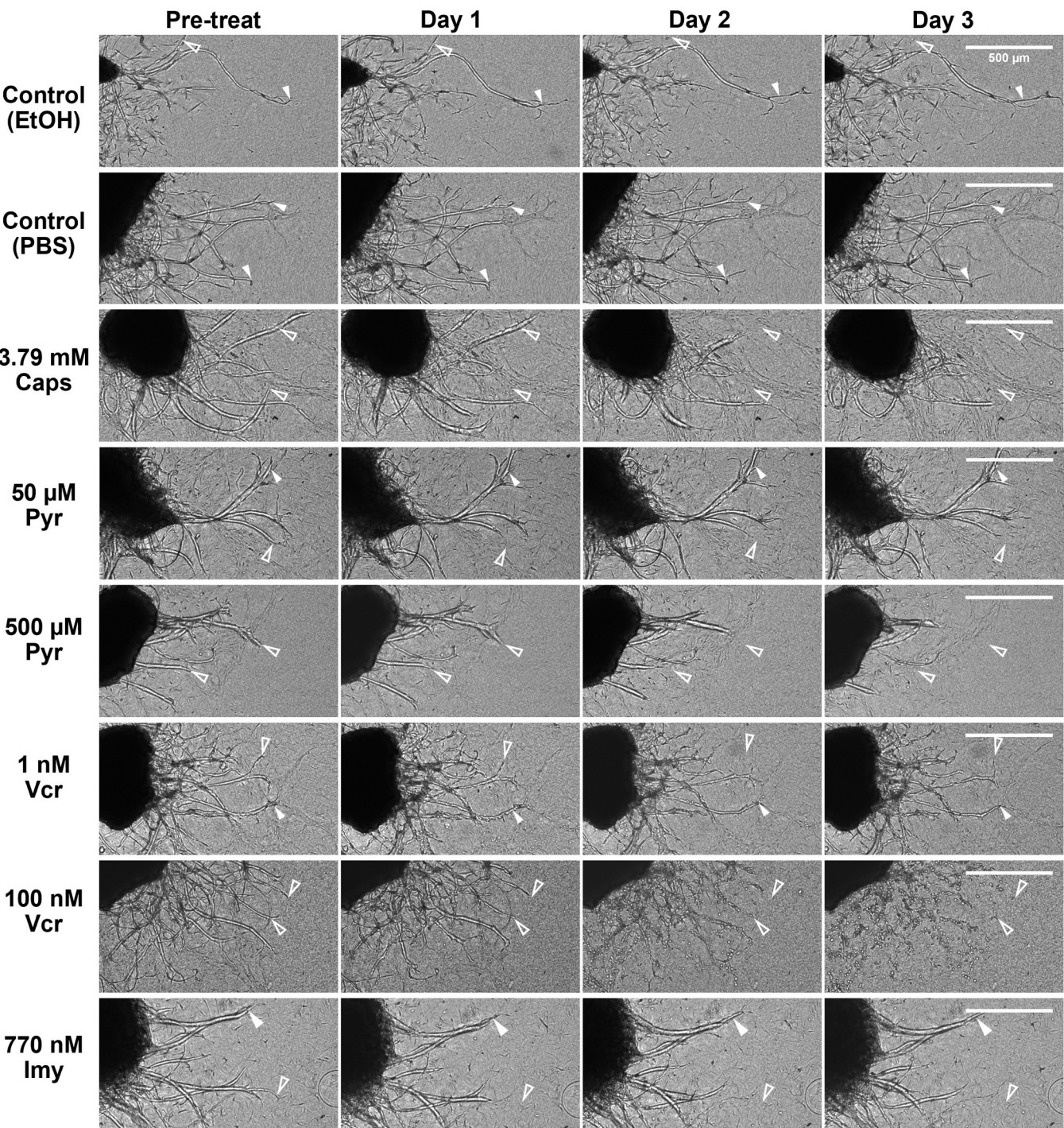

**Fig 1. DRG explants cultured in hydrogels have axon outgrowth in solvent controls but dieback after Caps, Pyr, Vcr and Imy.** Representative brightfield images of female rat DRGs show, over time, axons continue to extend (solid arrowheads) from day 1 to day 3 in control media with EtOH or PBS solvent and 50 μM Pyr, 1 nM Vcr, and 770 nM Imy. DRGs treated with 3.79 mM Caps, 500 μM Pyr and 100 nM Vcr have robust axonal dieback towards the soma (hollow arrowheads). 50 μM Pyr, 1 nM Vcr and 770 nM Imy have mixed axonal dieback and outgrowth. Scale bar: 500 μm. EtOH: ethanol, Pyr: Pyridoxine, Vcr: Vincristine sulfate, Imy: Ionomycin.

for analysis. DRG cytotoxicity in control and treatment groups were minimal compared to the positive control, lysed DRGs.

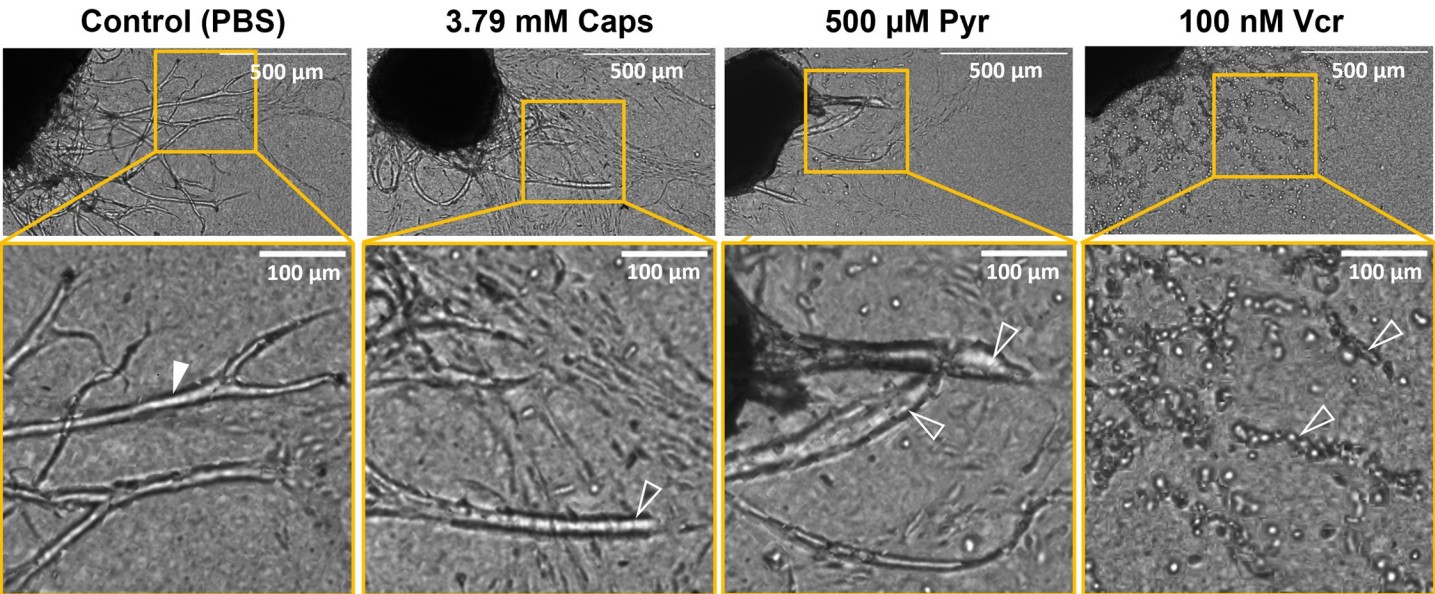

**Fig 2. DRG axon morphology was altered after treatment with capsaicin, pyridoxine and vincristine.** Representative brightfield images of female rat DRGs three days post-treatment show DRG axons treated with 3.79 mM Caps, 500 μM Pyr and 100 nM Vcr have altered axon morphology. Healthy DRG axons in PBS control group have smooth edges and continuous paths (solid arrowhead). Caps treated DRGs have axons with rounded ends without blebbing (clear arrowhead). 500 μM Pyr induced axonal dieback towards the soma and axonal swelling resulting in rounded nerve endings and thicker axons (hollow arrowheads). 100 nM Vcr induced signs of axonal degeneration as represented by axonal fragmentation, blebbing and beading in all axon paths (hollow arrowheads). Yellow boxes zooms into regions of interest. Pyr: Pyridoxine, Vcr: Vincristine sulfate.

## Pyridoxine induced axonal dieback in DRG explants in a dose-dependent manner without neurotoxicity but not sex-dependent manner

Pyr induces significant axonal dieback in DRGs from both sexes treated with concentrations of 500 μM compared to PBS solvent controls (**Figs 3C and S2A**). Pyr-induced axonal dieback by retracting axons towards the soma with blunted ends comparable to Caps-induced axonal dieback (**Figs 1 and 2 and S5 and S6 Videos**). 500 μM Pyr significantly reduced axon length ratio by day 3 compared to day 1 in male and female rat DRGs (p<0.001) indicating Pyr as a potential axonal dieback compound (**Fig 3C**). Further, compared to PBS solvent group, axon length ratio of the 500 μM Pyr group were significantly lowered on days 2 and 3 in male rat DRGs (p<0.001) and on day 3 in female rat DRGs (p = 0.01) (**Fig 3C**). Median axon length ratio of DRGs treated with 500 μM Pyr were close to Caps axon length ratio threshold on day 3 at 0.607 (0.418–0.875) and 0.539 (0.317–0.676) in male and female rat DRGs, respectively. At a lower dose, 50 μM Pyr did not induce significant axonal dieback over time and against solvent control in either male or female DRGs suggesting Pyr axonal dieback is dose dependent (**Fig 3C and S4 Video**). Treatment of DRGs with 50 and 500 μM Pyr did not cause a significant increase in percentage of cytotoxicity compared to PBS solvent control suggesting Pyr induces axonal dieback without neuronal cell death (**Fig 3D**).

Higher dose of Pyr at 1 mM in female rat DRGs significantly reduced axon length ratio compared to PBS control on day 1 (p = 0.005), day 2 (p<0.001) and day 3 (p<0.001) (**S2A Fig**). Also, axon length ratio significantly decreased between days 1, 2 and 3 (p<0.001) with 1 mM Pyr treatment (**S2A Fig and S6 Video**). The median axon length ratio of 1 mM Pyr reached the Caps axon length ratio threshold by day 2 at 0.451 (0.347–0.587) (**S2A Fig**). The percentage of cytotoxicity in 1 mM Pyr was similar to percentage of cytotoxicity at lower Pyr doses and was not significant compared to PBS control (**S2B Fig and Table 1**). No significant

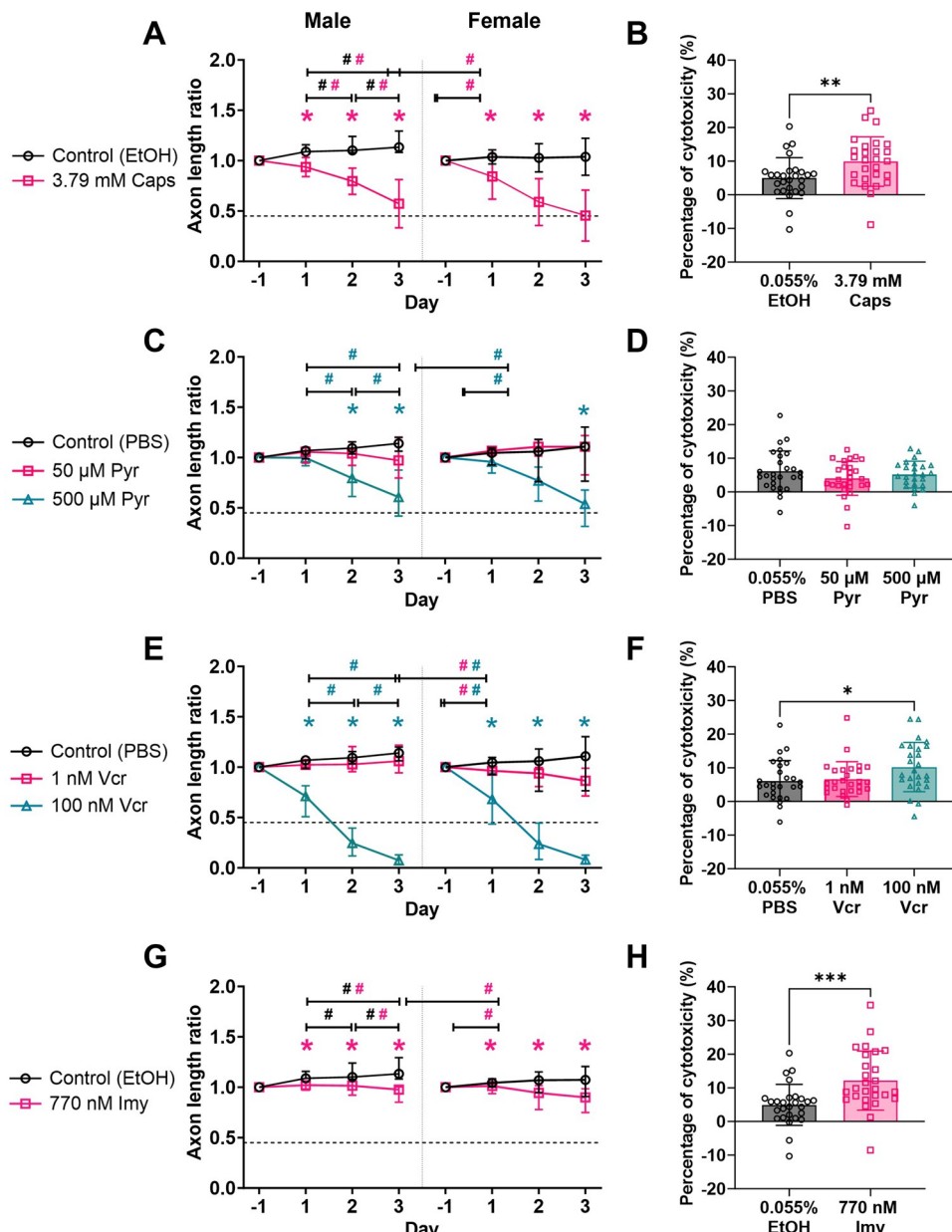

**Fig 3. Capsaicin, pyridoxine, vincristine sulfate and ionomycin significantly decreased axon length ratio but only pyridoxine had no impact on neurotoxicity.** DRG axon length ratio to pre-treat measured on days 1, 2 and 3 after axonal dieback compound screening with (**A**) Caps, (**C**) Pyr, (**E**) Vcr and (**G**) Imy in male (left) and female (right) rat DRG explants were compared over time (#$p < 0.05$) and between respective solvent controls (∗$p < 0.05$). (**A**) Caps significantly reduced axon length ratio over time and was significantly lower compared to EtOH control in both male and female rat DRGs. (**C**) Pyr at 500 µM significantly decreased axon length ratio over time and was significantly lower than PBS control by day 3. At a lower dose, 50 µM Pyr, did not induce significant change in axon length ratio over time and between PBS control in both female and male rat DRGs. (**E**) Vcr at 100 nM induced significant reduction in axon length ratio compared to PBS control and over time on days 1, 2 and 3. At a lower dose, 1 nM Vcr caused slow axonal dieback in female rat DRGs but did not impact male rat DRGs. (**G**) Imy at 770 nM induced significant reduction in axon length ratio over time and axon length ratio. Although, axon length ratio was significantly lower than EtOH control across all days, 770 nM Imy axonal dieback is very slow. (**B, D, F, H**) LDH assay revealed that Caps, 100 nM Vcr and 770 nM Imy induced significant increase in percentage of cytotoxicity compared to respective solvent control. Axon length ratio data shows the median and interquartile range of DRG replicates tested per group. Cytotoxicity data shows a scatter plot of each DRG replicate pooled from male and female rats with bar graph centered to the mean and error bars representing standard deviation. EtOH: ethanol, Imy: Ionomycin, Pyr: pyridoxine, Vcr: vincristine sulfate.

**Table 1. DRG cytotoxicity in rat DRG explants after axonal dieback screening.**

| Group | Mean (SD) | n (female, male rat DRGs) |
|---|---|---|
| 0.055% v/v EtOH | 4.92 (6.09) | 27 (13, 14) |
| 0.055% v/v 1X PBS | 6.12 (6.02) | 27 (13, 14) |
| 3.79 mM Caps | 9.85 (7.36) | 27 (13, 14) |
| 50 μM Pyr | 3.99 (5.01) | 27 (13, 14) |
| 500 μM Pyr | 5.13 (3.98) | 25 (11, 14) |
| 1 mM Pyr | 3.59 (4.61) | 12 (12, 0) |
| 500 nM Imy | 9.19 (5.19) | 14 (0, 14) |
| 770 nM Imy | 12.1 (8.78) | 27 (13, 14) |
| 1 nM Vcr | 6.64 (5.16) | 27 (13, 14) |
| 10 nM Vcr | 7.02 (4.36) | 14 (0, 14) |
| 100 nM Vcr | 10.2 (7.32) | 26 (12, 14) |
| 200 nM Vcr | 11.1 (9.78) | 12 (12, 0) |
| 500 nM Vcr | 12.0 (8.23) | 12 (12, 0) |
| Lysed | 100 (51.6) | 27 (13, 14) |

Mean and standard deviation of percentage of DRG cytotoxicity from male and female rat DRGs after treatment with either vehicle control or compound solution. The percentage of cytotoxicity did not differ significantly between male and female rat DRGs and were pooled for analysis. DRG cytotoxicity in control and treatment groups were minimal compared to the positive control, lysed DRGs.

sex differences in axon length ratio and neurotoxicity were detected in Pyr-treated DRGs. Overall, our results indicate Pyr does not induce DRG apoptosis at any dose and high doses of Pyr above 500 μM induce axonal dieback of DRG axons within three days.

## Vincristine sulfate induced axonal dieback in DRG explants in a dose-dependent manner that is not sex-dependent and minimal neurotoxicity

Vcr induced robust and significant axonal dieback compared to PBS solvent controls by day 3 after treatment with 100 nM but not at 1 nM consistently in both male and female rat DRGs (**Fig 3E**). This result suggests Vcr-induced axonal dieback is dose-dependent and not sex-dependent. Vcr-treated DRGs at 100 nM retracted and underwent fragmentation and beading on day 1 before completely disappearing and leaving behind cellular debris on day 3 (**Figs 1 and 2 and S8 Video**). Median axon length ratio in 100 nM Vcr reduced significantly between days 1, 2 and 3 and eventually reached 0.081 (0.039–0.125) in female rat DRGs (p = 0.002) and 0.074 (0.037–0.129) in male rat DRGs (p<0.001) by day 3 (**Fig 3E**). These data indicate that less than 8.1% of the original axon length was left behind after 100 nM Vcr which is substantially lower compared to Caps and Pyr-treated DRGs. 100 nM Vcr met Caps axon length ratio threshold by day 2 post-treatment. An intermediate dose of 10 nM Vcr also induced significant axonal dieback in male rat DRGs compared to solvent controls, but axonal dieback was not as robust as 100 nM Vcr since the axon length ratio only reached 0.613 (0.430–0.723) on day 3 (**S2E Fig**). Interestingly, at a lower dose, the axon length ratio after 1 nM Vcr treatment was sex-dependent (**Fig 3E**). In the 1 nM Vcr group, male rat DRG axon length ratio increased to 1.06 (0.944–1.217) on day 3, but female rat DRGs were more sensitive, and the ratio significantly decreased compared to day 1 (p = 0.02) reaching 0.866 (0.714–0.989) by day 3 (**Fig 3E and S7 Video**). Although axon growth was observed in male rat DRGs and dieback was observed in female DRGs after 1 nM Vcr treatment, both axon length ratios were not significant compared to PBS control indicating 1 nM Vcr is not a robust axonal dieback compound

at low doses (**Fig 3E**). Treatment of DRGs with 1 nM Vcr did not result in significant neurotoxicity compared to solvent control; however, 100 nM Vcr did cause a significant increase in neurotoxicity (p = 0.03) (**Fig 3F**). Nevertheless, the percentage of cytotoxicity is minimal and significantly lower, 89.8% less, than lysed DRGs (**S3 Fig** and **Table 1**).

Higher doses of Vcr tested on female rat DRGs had significant decrease in axon length ratios between days 1 and 3 after incubation with 200 nM Vcr (p<0.001) and 500 nM Vcr (p<0.001) (**S2C Fig** and **S9, 10 Videos**). 200 nM and 500 nM Vcr-treated DRGs had significantly lower axon length ratio than PBS control on days 1, 2 and 3 (p<0.001) and is not neurotoxic (**S2C and S2D Fig**). In comparison to Caps, 200 and 500 nM Vcr met the axon length threshold by day 1. The percentage of cytotoxicity in 200 nM and 500 nM Vcr cytotoxicity were not significantly different compared to PBS control (**S2D Fig**). Overall, these results suggest that axonal dieback was dose-dependent where high doses Vcr above 100 nM was a robust axonal dieback compound with minimal neurotoxicity.

## Ionomycin induced slow axonal dieback in DRG explants and minimal neurotoxicity

Testing the maximum available dose on male and female DRGs at 770 nM Imy, there are only a few axons that exhibit robust axonal dieback after 770 nM Imy treatment and the majority of DRG axons were still present on day 3 (**Fig 1**). The axon length ratio of the DRGs treated with 770 nM Imy had a small decrease but significantly reduced over time and were significantly lower compared to EtOH controls on days 1, 2 and 3 in both male (p = 0.004) and female rat DRGs (p = 0.02) (**Fig 3G**). The median axon length ratios for 770 nM Imy group were not sex-dependent and reached 0.977 (0.852–1.004) in male rat DRGs and 0.9 (0.75–0.985) in female rat DRGs on day 3 indicating the axons only had 10% decrease in length, which is insufficient to meet the Caps axon length threshold. The percentage of cytotoxicity was significantly increased compared to EtOH control but remained significantly lower compared to lysed DRGs (**Figs 3H** and **S3** and **Table 1**). No sex differences were detected in axon length ratio and DRG cytotoxicity after 770 nM Imy treatment (**Fig 3G**). Comparable results were obtained when male rat DRGs were treated with 500 nM Imy; the median axon length ratio decreased slightly to 0.982 (0.854–1.121) on day 3 after treatment. Axon length ratios of DRGs treated with 500 nM Imy were significantly lower compared to EtOH controls on days 1, 2 and 3, but no significant differences were observed between timepoints indicating Imy inhibited axonal outgrowth (**S2G Fig**). 500 nM Imy did not cause significant DRG cytotoxicity compared to EtOH solvent controls (**S2H Fig**). Overall, 500 nM and 770 nM Imy have minimal neurotoxicity but did not cause robust axonal retraction.

## Pyridoxine and vincristine sulfate are not cytotoxic to human nucleus pulposus cells

Pyr induced robust axonal dieback at doses at or above 500 µM in DRG explants. Hence, we tested the cytocompatibility of 500 µM, 1 mM and 2 mM Pyr on human NP cells in vitro. NP cells remained viable and proliferated after exposure to Pyr at all doses (**Fig 4A and 4C**). The 2 mM Pyr group had slightly lower cell confluency suggesting a slower proliferation rate, which also resulted in significantly reduced cell viability compared to PBS control in donor 2, and larger variation in metabolic activity in both donors (**Fig 4A–4C**). Nevertheless, the percentage of viable cells were greater than 89% by day 3 post-treatment in all Pyr groups in either donor (**Fig 4A**). NP ells treated with 500 µM, 1 mM and 2 mM Pyr stayed metabolically active as AlamarBlue reduction normalized to cell counts were not significantly different between PBS control and all treated groups (**Fig 4B**).

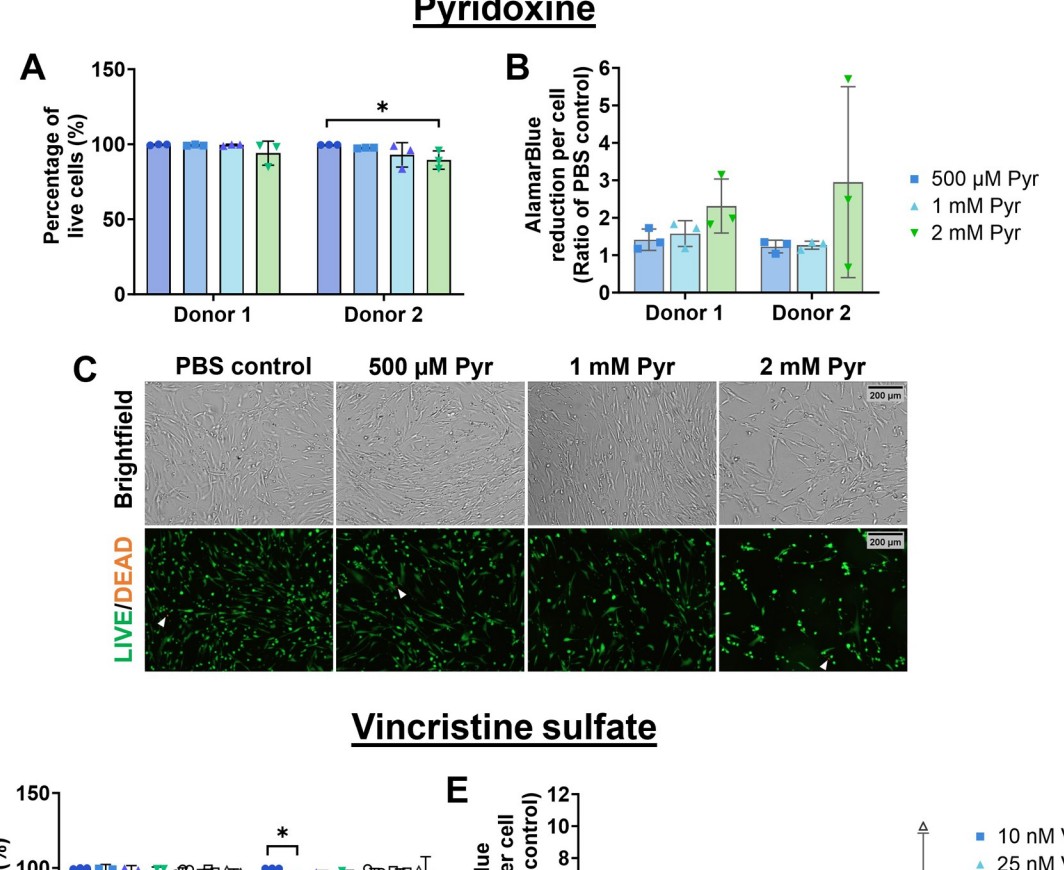

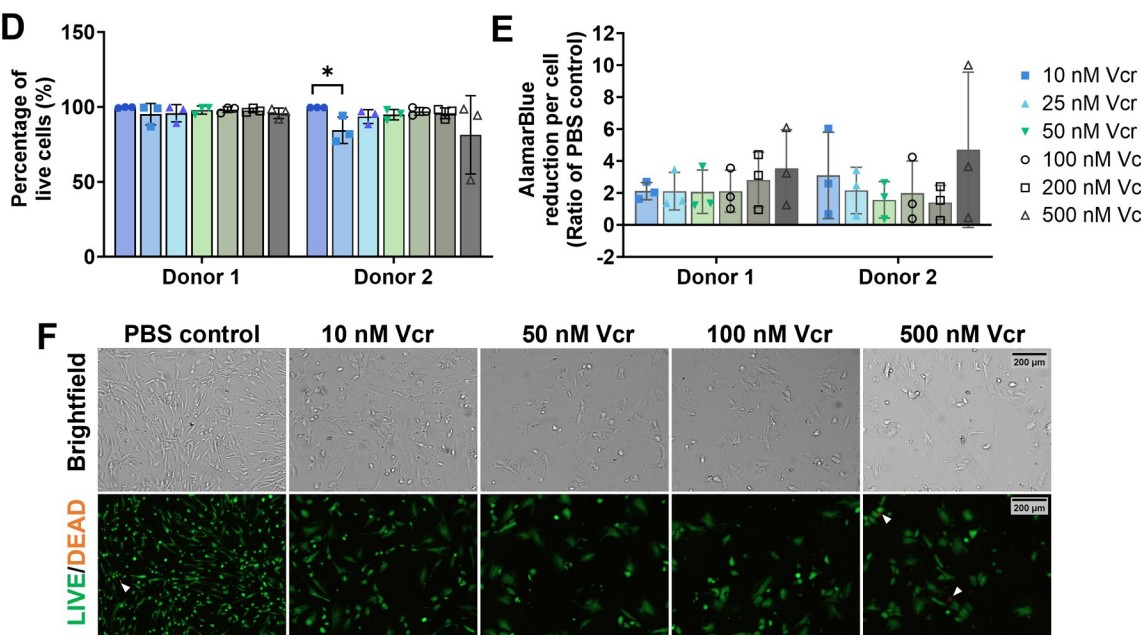

**Fig 4. Pyridoxine and vincristine sulfate are not cytotoxic to human NP cells in vitro after 72-hour incubation.** Pyr (500 μM, 1mM) did not have significant effect on (**A**) cellular viability and (**B**) metabolic activity in either donor. (**C**) Representative brightfield images show normal morphology after 72 hours incubation with 500 μM, 1 mM and 2 mM Pyr. Cell morphology after LIVE/DEAD staining (bottom) was rounder compared to non-stained NP cells (top). LIVE/DEAD staining showed human NP cells remained viable after Pyr treatment as indicated by positive green (live) cell staining and very limited positive orange cells (dead). Vcr (25–500 nM) did not have a significant effect on (**D**) cell viability and (**E**) metabolic activity in either donor. (**F**) Representative brightfield images after treatment with various Vcr concentrations. Vcr-treated cells have rounder morphology and tend to aggregate. NP cells did not proliferate after Vcr treatment compared to PBS controls. Nevertheless, cells remained viable according to positively stained green cells after LIVE/DEAD staining (bottom) with few dead cells. White arrows point to dead cells. Bar graphs represent the mean and standard deviation from n = 3 experimental replicates of two different donors. Scale bar: 200 μm. Pyr: Pyridoxine, Vcr: Vincristine sulfate.

Vcr at high doses at or above 10 nM also induced robust axonal dieback in DRG explants. Low dose Vcr at 1 nM did not induce axonal dieback; thus, we decided to evaluate the cyto-compatibility of Vcr doses ranging between 10 nM to 500 nM on human NP cells in vitro. No significant differences were detected in percentage of viable cells and cellular metabolic activity in cells from both donors in NP cells treated with Vcr ranging between 10 to 500 nM compared to PBS control in donor 1 and 2 except 10 nM Vcr significantly reduced NP cell viability in donor 2 (p = 0.03). Despite 10 nM Vcr significantly reducing NP cell viability, cell metabolic activity of cells was not affected in both donors. In general, Vcr did not induce significant cell death or change in cell metabolic activity compared to PBS solvent controls (**Fig 4D and 4E**). Vcr-treated NP cells did not proliferate as much as control non-treated groups as evident by lower confluency in all Vcr-treated groups (**Fig 4F**). Vcr-treated NP cells also exhibited cellular aggregates and a larger and slightly rounder cell morphology compared to PBS control NP cells (**Fig 4F**). Even though cell morphology and counts were altered in the Vcr groups, NP cells remain viable (greater than 81% viable cells) by day 3 post-treat (**Fig 4D**) suggesting Vcr does not induce substantial cell death of human NP cells. There is a significant donor effect in percentage of live cells (p<0.001) but not in cell metabolic activity. These results demonstrated that Pyr within 500 μM to 1 mM and Vcr within 25 nM to 500 nM does not induce significant cell death or affect cell metabolic activity and thus, are cytocompatible with human NP cells.

## Discussion

Using an adult rat DRG explant hydrogel culture platform, we observed similar axon out-growth length and rates between male and female rat DRGs (**S1 Fig**) and significant axonal dieback within three days after treating DRGs with 3.79 mM Caps, 500 μM and 1 mM Pyr, and 10, 100, 200, 500 nM Vcr (**Figs 3 and S2**). Caps confirmed the use of our DRG explant culture platform as a tool for screening axonal dieback compounds. Our findings are the first to demonstrate axonal dieback of Pyr, Vcr, and Imy on adult rat DRG neurons. Pyr and Vcr induced axonal dieback in a dose-dependent but not DRG sex-dependent manner. DRGs treated with 770 nM Imy had slow axonal dieback and 500 nM Imy inhibited axonal outgrowth. DRGs treated with any dieback compound at the range of concentrations tested had substantially lower levels of cytotoxicity compared to lysed DRGs suggesting neuronal cell death is minimal (**S3 Fig**). We also showed that Pyr and Vcr within concentration ranges that induced axonal dieback are cytocompatible with human NP cells (**Fig 4**). In summary, Pyr and Vcr hold excellent potential as axonal dieback compounds for local denervation.

Geisler et al demonstrated 40 nM Vcr induced axon degeneration within 12 hours and the axons completely fragmented by 36 hours in dissociated DRG neurons from embryonic mice [50]. We observed similar varicosity formation and fragmentation at approximately 48 hours or day 2 post-treat in adult rat DRG explant axons treated with 100–500 nM Vcr (**Figs 1 and 2 and S8–S10 Video**). The degenerated DRG axons after Vcr treatment in our study were also identical to transected axons in an in vitro model of Wallerian degeneration using superior cervical ganglia [97]. Hence, based on the axon morphology and reduction in axon length combined with its limited neurotoxicity, Vcr is a suitable axon dieback compound for denervation.

On the other hand, our DRG axons treated with 500 μM to 1 mM Pyr retracted towards the soma without leaving behind any debris or forming varicosities in our DRG explant model (**Fig 2 and S9 and S10 Videos**). One potential mechanism of Pyr-induced nerve degeneration is the disruption of gamma-aminobutyric acid (GABA) signaling causing excess depolarization of neurons and calcium release which can lead to axonal degeneration [98]. Previous studies have observed changes in axon morphology such as axonal swelling [67], vacuolation [67] and

segmental demyelination [48] after testing Pyr in vivo but we did not observe the same changes in our Pyr-treated adult rat DRG explant neurons here.

Imy at 500 nM and 770 nM did not have robust axonal dieback in our study. In a different neuronal cell type, Nakamura et. al induced a dose and time-dependent response axon degeneration and cell death in mouse neuroblastoma cells with Imy [59]. Longer incubation times and higher concentrations of Imy could be tested but would require increasing the amount of EtOH in the treatment solution to attain high doses. This high EtOH concentration must be carefully considered since high EtOH exposure can also cause axonal degeneration in rats [99].

Sex differences were not observed in axonal dieback in all doses of Pyr, Imy and at high dose Vcr groups (**Fig 3**). However, female rat DRGs were more susceptible to Caps and 1 nM Vcr than male rat DRGs. We are the first to show Caps-induced axon dieback is different by sex of rat DRGs used. Our results provide evidence that axonal dieback induced by Pyr, Vcr and Imy is independent of sex of rat at the doses tested herein.

An important consideration for selecting an axonal dieback compound is DRG apoptosis as neuronal cell death is highly associated with pain [83]. Our findings demonstrated that Pyr is not neurotoxic (**Fig 3D**) but 3.79 mM Caps, 100 nM Vcr and 770 nM Imy had significantly increased neurotoxicity compared to matched solvent controls (**Figs 3 and S2**). However, the percentage of cytotoxicity across all groups was less than 12.1% and was significantly lower compared to lysed DRGs. Our results verified the findings from Kramer et al that determined Vcr below 1000 nM does not induce cytotoxicity in adult rat DRG neurons [100]. Chard et al. found 30 μM Caps, which is 126-fold lower than our study, induced significant cell death in dissociated neonatal DRG neurons within 24 hours [101]. Previous studies in neonatal mice [102] and rats [101] also observed a transient increase in apoptotic cells that peaked at 24 hours after direct Caps injection. Therefore, it is unsurprising to see a significant increase in neurotoxicity after Caps treatment in our study. Previous study with mouse cortical neuron culture showed LDH release and slow neuronal death with Imy which is also consistent with our findings [103]. Imy also triggered apoptotic cell death in rat retinal ganglion cells treated with Imy via calpain activation [104]. Our results were the first to observe the neurotoxic effects of Imy on adult rat DRG explants. In addition, we are also the first to demonstrate that Pyr is not neurotoxic to DRG neurons which revealed that neuronal cell death is not associated to Pyr-induced axonal dieback. Our data provides preliminary evidence that Pyr, Vcr and Imy will not induce high neurotoxicity and future in vivo studies to confirm the safety of these compounds is needed.

In our study, Pyr and Vcr had the most robust axonal dieback in adult rat DRG explants and demonstrated maintenance of cell viability and metabolic activity after 3-days incubation in human NP cells (**Fig 4**). Therefore, Pyr and Vcr are non-cytotoxic to NP cells and can potentially be repurposed as axonal dieback compounds for local denervation in intervertebral discs. Pyr between 500 μM to 2 mM did not affect NP cell viability, metabolism, shape, or proliferation (**Fig 4A–4C**). However, NP cells aggregated, altered cell morphology and reduced proliferation after treatment with 10 nM to 500 nM Vcr (**Fig 4D–4F**) which may be explained by high affinity of Vcr to β-tubulin which is a major cytoskeletal cell component important for cell division [69]. Albeit Vcr blocking NP cell proliferation, NP cells were still viable and stayed metabolically active (**Fig 4D and 4E**). Cytocompatibility testing in other cell types such as keratinocytes, chondrocytes, and epithelial cells, would be necessary for denervation applications in other target tissues such as neuromas, knee joints and tumors.

A limitation of our DRG explant culture platform is that the neuronal cell bodies in the soma and axons were both exposed simultaneously to the compounds during testing. This set up is not physiologically relevant as axonal dieback compounds for denervation will need to be

delivered locally to target nerve ending sites where aberrant nerve sprouting is occurring. Multicompartment systems for adult rat DRG explant culture can be used to test the effects of axonal dieback on distal axons away from the soma [70]. Another limitation of this study is the limited observation period. Longer observation periods to check reversibility of axonal dieback may be useful to predict the long-term effectiveness of denervation. Previously published work that have developed neuroinhibitory biomaterials that could be used as a co-treatment to potentially curb nerve regrowth after denervation for potentially longer lasting pain relief [89, 90, 105, 106]. Future work to investigate pain alleviation efficacy using in vivo models must be performed to investigate if robust axonal dieback induced by high dose Pyr and Vcr can repurposed these compounds as pain relief drugs for chronic pain conditions with aberrant nerve sprouting.

In conclusion, we have demonstrated herein that adult rat DRG explant hydrogel culture platforms can be used to identify and characterize neurotoxicity of axonal dieback compounds. We provided first evidence that Pyr and Vcr induces dose-dependent axonal degeneration in adult male and female rat DRG explants without major concerns for neurotoxicity and are cytocompatible with human NP cells in vitro.

## Supporting information

**S1 Data. Raw data of DRG axon length and cytotoxicity data.**
(XLSX)

**S2 Data. Raw data of nucleus pulposus cell viability and metabolic activity.**
(XLSX)

**S1 Fig. DRGs harvested from adult male and female rats had no significant sex differences in axon length and growth rate in 3D hydrogel in vitro culture platform.** (**A**) Traced axon lengths and of male rat DRGs and female rat DRGs measured on days 7, 14 and 20 increased significantly over time. No significant differences were detected in traced axon length between male and female DRGs within each day. (**B**) Similarly, maximum radial distance of male rat DRGs and female rat DRGs increased significantly from days 7, 14 to 20, but no sex differences were detected. (**C**) Average axon growth rate calculated as the percent difference of traced axon length on day 14 to day 7 and on day 20 to day 14, respectively showed both male and female rat DRGs have significantly higher growth rate between day 7 to 14 compared to growth rate from day 14 to 20. Sex differences in DRG axon growth rate was not detected. (TIF)

**S2 Fig. Additional doses of pyridoxine, vincristine sulfate and ionomycin tested on female and male rat DRGs induced axonal dieback without significant neurotoxicity.** At higher doses, axon length ratio significantly decreased with (**A**) 1 mM Pyr, (**C**) 200 nM Vcr and 500 nM Vcr compared to matched solvent controls in female rat DRGs as expected. Testing lower doses, (**E**) 10 nM Vcr and (**G**) 500 nM Imy also had significantly lowered axon length ratio compared to matched controls. However, 500 nM Imy did induce significant change in axon length ratio over time which suggests 500 nM Imy prevents axonal outgrowth without dieback. (**B, D, F, H**) LDH assay did not detect any significance in percentage of cytotoxicity levels of 1 mM Pyr, 200 nM Vcr, 500 nM Vcr and 500 nM Imy. The asterisk (*) symbol represents significant difference in axon length ratio compared to control on each day while the hashtag (#) symbol represents significant difference in axon length ratio between days. Axon length ratio data shows the median and interquartile range of DRG replicates tested per group. Cytotoxicity data shows a scatter plot of each DRG replicate with bar graph centered to the mean and error bars representing standard deviation. EtOH: ethanol, Imy: Ionomycin, Pyr: pyridoxine,

Vcr: vincristine sulfate.
(TIF)

**S3 Fig. No significant sex differences were detected in percentage of cytotoxicity levels.** All DRGs treated with solvent control and compounds of interest have significantly lower percentage of cytotoxicity compared to lysed DRGs across all concentrations. The scatter plot shows value from DRG replicate and mean and standard deviation.
(TIF)

**S4 Fig. Dead NP cells stained positive orange from LIVE/DEAD staining.** Human NP cells were incubated with media for three days. Then, cells were lysed with 70% Ethanol for 10 minutes at 37˚C, stained with LIVE/DEAD staining solution for 30 minutes at room temperature then rinsed with 1X PBS and imaged using Cytation plate imager (Agilent). This well was used as a positive control for dead cells and to determine the image settings for the RFP channel.
(TIF)

**S1 Video. Axon morphology of EtOH treated DRG axons.** Series of brightfield images before and after on days 1, 2 and 3 post-treatment show axonal growth and normal morphology with smooth, continuous axonal paths in EtOH vehicle control group.
(GIF)

**S2 Video. Axon morphology of PBS treated DRG axons.** Series of brightfield images before and after on days 1, 2 and 3 post-treatment show axonal growth and normal morphology with smooth, continuous axonal paths in PBS vehicle control group.
(GIF)

**S3 Video. Axonal morphology of capsaicin treated DRG axons.** Series of brightfield images before and after on days 1, 2 and 3 post-treatment with 3.79 mM Caps show progression of axonal dieback towards the soma with rounded nerve endings.
(GIF)

**S4 Video. Axonal morphology of pyridoxine (50 μM Pyr) treated DRG axons.** Series of brightfield images before and after on days 1, 2 and 3 post-treatment show normal axonal morphology and growth when treated with 50 μM Pyr.
(GIF)

**S5 Video. Axonal morphology of pyridoxine (500 μM Pyr) treated DRG axons.** Series of brightfield images before and after on days 1, 2 and 3 post-treatment show progression of axonal dieback towards the soma with rounded nerve endings when treated with 500 μM Pyr. Axons appear thicker and have bulbed endings.
(GIF)

**S6 Video. Axonal morphology of pyridoxine (1 mM Pyr) treated DRG axons.** Series of brightfield images before and after on days 1, 2 and 3 post-treatment show progression of axonal dieback towards the soma with rounded nerve endings when treated with 1 mM Pyr. Axons completely disappear on day 3 post-treatment.
(GIF)

**S7 Video. Axonal morphology of pyridoxine (1 nM Vcr) treated DRG axons.** Series of brightfield images before and after on days 1, 2 and 3 post-treatment show progression of axonal beading but no reduction in axonal length or presence of fragmentation when when treated with 1 nM Vcr.
(GIF)

**S8 Video. Axonal morphology of pyridoxine (100 nM Vcr) treated DRG axons.** Series of brightfield images before and after on days 1, 2 and 3 post-treatment show progression of axonal dieback towards the soma with axonal fragmentation when treated with 100 nM Vcr. (GIF)

**S9 Video. Axonal morphology of pyridoxine (200 nM Vcr) treated DRG axons.** Series of brightfield images before and after on days 1, 2 and 3 post-treatment show progression of axonal dieback towards the soma with axonal fragmentation when treated with 200 nM Vcr. (GIF)

**S10 Video. Axonal morphology of pyridoxine (500 nM Vcr) treated DRG axons.** Series of brightfield images before and after on days 1, 2 and 3 post-treatment show progression of axonal dieback towards the soma with axonal fragmentation when treated with 500 nM Vcr. (GIF)

## Author Contributions

**Conceptualization:** Fei San Lee.

**Data curation:** Fei San Lee, Uyen N. Nguyen, Eliza J. Munns, Rebecca A. Wachs.

**Formal analysis:** Fei San Lee, Uyen N. Nguyen, Eliza J. Munns, Rebecca A. Wachs.

**Funding acquisition:** Rebecca A. Wachs.

**Investigation:** Fei San Lee, Rebecca A. Wachs.

**Methodology:** Fei San Lee, Rebecca A. Wachs.

**Supervision:** Rebecca A. Wachs.

**Writing – original draft:** Fei San Lee.

**Writing – review & editing:** Fei San Lee, Uyen N. Nguyen, Eliza J. Munns, Rebecca A. Wachs.

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
