## [Decision Letter · Decision Letter 0]

20 Nov 2023

PONE-D-23-27605Identification of compounds that cause axonal dieback without cytotoxicity in dorsal root ganglia explants and intervertebral disc cells with potential to treat pain via denervationPLOS ONE

Dear Dr. Wachs,

Thank you for submitting your manuscript to PLOS ONE. After careful consideration, we feel that it has merit but does not fully meet PLOS ONE’s publication criteria as it currently stands. Therefore, we invite you to submit a revised version of the manuscript that addresses the points raised during the review process. Both reviewers view your manuscript positively. However, Reviewer 2, in particular, makes detailed criticisms that you are requested to address as fully as possible in your revised manuscript.

We look forward to receiving your revised manuscript.

Kind regards,

Israel Silman

Academic Editor

PLOS ONE

3. Please include your tables as part of your main manuscript and remove the individual files. Please note that supplementary tables (should remain/ be uploaded) as separate "supporting information" files.

“RAW received the NSF CAREER award (NSF CAREER1846857) from the National Science Foundation (NSF) which provided funds to conduct this research. RAW, FSL, UNN received funding by NSF CAREER1846857. EJM received funding from NSF Research Experiences for Undergraduates (REU) program EEC 2050587 as a part of this work.”

“We would like to thank the National Science Foundation (NSF) for providing funds to conduct this research (NSF CAREER1846857). We would also like to acknowledge the NSF Research Experiences for Undergraduates (REU) program EEC 2050587 for supporting part of this work.”

“RAW received the NSF CAREER award (NSF CAREER1846857) from the National Science Foundation (NSF) which provided funds to conduct this research. RAW, FSL, UNN received funding by NSF CAREER1846857. EJM received funding from NSF Research Experiences for Undergraduates (REU) program EEC 2050587 as a part of this work.”

6. We note that you have indicated that data from this study are available upon request. PLOS only allows data to be available upon request if there are legal or ethical restrictions on sharing data publicly. For more information on unacceptable data access restrictions, please see http://journals.plos.org/plosone/s/data-availability#loc-unacceptable-data-access-restrictions.

7. Please include a separate caption for each figure in your manuscript.

8. We note that Figure 1 in your submission contain copyrighted images. All PLOS content is published under the Creative Commons Attribution License (CC BY 4.0), which means that the manuscript, images, and Supporting Information files will be freely available online, and any third party is permitted to access, download, copy, distribute, and use these materials in any way, even commercially, with proper attribution. For more information, see our copyright guidelines: http://journals.plos.org/plosone/s/licenses-and-copyright.

10. We notice that your supplementary figures are uploaded with the file type 'Figure'. Please amend the file type to 'Supporting Information'. Please ensure that each Supporting Information file has a legend listed in the manuscript after the references list.

Reviewers' comments:

Reviewer's Responses to Questions

**Comments to the Author**

1. Is the manuscript technically sound, and do the data support the conclusions?

Reviewer #1: Yes

Reviewer #2: Yes

2. Has the statistical analysis been performed appropriately and rigorously? 

Reviewer #1: Yes

Reviewer #2: Yes

3. Have the authors made all data underlying the findings in their manuscript fully available?

Reviewer #1: Yes

Reviewer #2: Yes

4. Is the manuscript presented in an intelligible fashion and written in standard English?

Reviewer #1: Yes

Reviewer #2: Yes

5. Review Comments to the Author

Reviewer #1: The work describes the effects of pyridoxine (Pyr), vincristine sulfate (Vcr) and ionomycin (Imy) on DRG explant axonal dieback and cytotoxicity. The experiments and results have been described very clearly.

I have only two comments:

-As the Authors stated, the used DRG explant model has the limitation that both the neuronal cell bodies and the axons have been exposed to the tested compounds. I wonder whether this model is physiologically

relevant to study axonal dieback. While this first screening was important, the Authors should include an experiment where the compounds are tested on the axonal compartment only, to make their findings more physiologically relevant.

-All the analysis have been done on bright-field images. It would be meaningful complementing with fluorescence analysis, using axonal markers, to have a better understanding of the axon morphology (axon blebbing, axon retraction).

Reviewer #2: The manuscript investigates potential application of pyridoxine (Pyr) and vincristine sulfate (Vcr) to cause local denervation and alleviate pain induced by aberrant nerve growth in tissue. The author utilized rat DRG explant hydrogel culture platform for axonal dieback compound screening under four drugs: capsaicin (Caps), Pyr, Vcr, and ionomycin (Imy). Axonal length was measured before and after drug treatment to calculate axon length ratio. To assess cytotoxicity of the drugs, DRG lactate dehydrogenase (LDH) activity was measured post drug treatment. Drug toxicity to human nucleus pulposus (NP) cells was also studied in vitro through cell viability kits. Results showed that Pyr and Vcr was able to induce reduction in axonal length at high concentration while Imy produced slower axonal dieback. Imy, Vcr and Pyr had only minimal neurotoxicity. Further, neither Pyr nor Vcr triggered NP cell death or affected cellular metabolic activity. Overall, the research is exciting and brought up new axon dieback candidates to treat pain without induced cytotoxicity to intervertebral disc cells, but there are some concerns need to be addressed:

1. 3.79 mM Caps was used as positive control for the study, but there lacks detailed description how the concentration was calculated. Although 8% Caps is used in commercial patch, it’s better to investigate the effects of different concentration on DRG explant to strengthen the advantages of Pyr and Vcr over Caps.

2. The image quality of DRG outgrowth (Figure 2) is low and it’s hard to visualize the axons.

3. Could the author specify how many axons were included for each DRG explant piece for axonal length (ratio) measurement? Meanwhile, it’s unclear if the axonal length referred to the maximum axonal length or to the average length of several selected axons. If only several axons were selected, could the author clarify the criteria of selection? How did the author make sure the exact same axon was captured before and after drug treatment to calculate the axonal length change (ratio)?

4. The author may consider doing immunostaining of the neurites for higher resolution images and more precise quantification. For example, TUJ staining will allow the visualization of neurite outgrowth and Sholl analysis in ImageJ for average axonal length measurement.

5. Instead of having two separate figures, the reviewer would recommend showing the dose-dependent drug effect on axonal length and cytotoxicity in one figure, i.e., combining the Figure 2 and supplementary Figure 2 for clearer interpretation.

6. Could the author explain why they normalized the LDH activity of DRG explant to lysed DRGs? In addition, the term “neurotoxicity” has to be used carefully since DRG consists of various types of cells.

7. The AlamarBlue reduction per cell plot in Figure 4 is a bit confusing. Could the author explain why the replicates in PBS control group had the same value? Some groups seem to have significant higher values compared to control (for example 1mM Pyr vs PBS in Donor1) while not highlighted in the plot. In addition to the 2-way ANOVA test, multiple comparison against control is something the author should consider performing here.

6. PLOS authors have the option to publish the peer review history of their article (what does this mean?). If published, this will include your full peer review and any attached files.

Reviewer #1: No

Reviewer #2: **Yes: **Mikhail Berezin

---

## [Decision Letter · Decision Letter 1]

26 Feb 2024

Identification of compounds that cause axonal dieback without cytotoxicity in dorsal root ganglia explants and intervertebral disc cells with potential to treat pain via denervation

PONE-D-23-27605R1

Dear Dr. Wachs,

We’re pleased to inform you that your manuscript has been judged scientifically suitable for publication and will be formally accepted for publication once it meets all outstanding technical requirements.

Kind regards,

Israel Silman

Academic Editor

PLOS ONE

Additional Editor Comments (optional):

Reviewers' comments:

Reviewer's Responses to Questions

**Comments to the Author**

1. If the authors have adequately addressed your comments raised in a previous round of review and you feel that this manuscript is now acceptable for publication, you may indicate that here to bypass the “Comments to the Author” section, enter your conflict of interest statement in the “Confidential to Editor” section, and submit your "Accept" recommendation.

Reviewer #1: (No Response)

Reviewer #2: All comments have been addressed

2. Is the manuscript technically sound, and do the data support the conclusions?

Reviewer #1: Partly

Reviewer #2: Yes

3. Has the statistical analysis been performed appropriately and rigorously? 

Reviewer #1: Yes

Reviewer #2: Yes

4. Have the authors made all data underlying the findings in their manuscript fully available?

Reviewer #1: Yes

Reviewer #2: (No Response)

5. Is the manuscript presented in an intelligible fashion and written in standard English?

Reviewer #1: Yes

Reviewer #2: Yes

6. Review Comments to the Author

Reviewer #1: Thank you for the video and the high magnification pictures. They are much clearer.

Regarding my previous concern about the necessity of testing the compounds on the axon compartment, I do still believe it would be an important addition. With all the limitations of a 2D assay, it will add a valuable information and it will not be a duplication of the experiment but a valuable supplement. In fact, the Authors themselves write " There is a need to characterize the axonal degeneration effects when applying these drugs locally on nerve fiber endings to determine their potential efficacy to alleviate pain by local denervation without off-target effects".

Regarding the neurotoxicity vs cytotoxicity distinction: it would be important to perform the cytotoxicity assay on DRG neuron lysates. There are protocols to enrich the preparation for DRG neurons (for example: Tsantoulas et al., PLOSOne, 2013 https://doi.org/10.1371/journal.pone.0080722).

Reviewer #2: (No Response)

7. PLOS authors have the option to publish the peer review history of their article (what does this mean?). If published, this will include your full peer review and any attached files.

Reviewer #1: No

Reviewer #2: **Yes: **Mikhail Berezin

---

## [Editor Report · Acceptance letter]

13 Mar 2024

PONE-D-23-27605R1 

PLOS ONE

Dear Dr. Wachs, 

I'm pleased to inform you that your manuscript has been deemed suitable for publication in PLOS ONE. Congratulations! Your manuscript is now being handed over to our production team.

Kind regards, 

on behalf of

Prof. Israel Silman 

Academic Editor

PLOS ONE